# Chemical and Mechanical Properties of Metakaolin-Based Geopolymers with Waste Corundum Powder Resulting from Erosion Testing

**DOI:** 10.3390/polym14235091

**Published:** 2022-11-23

**Authors:** Giovanni Dal Poggetto, Pakamon Kittisayarm, Suphahud Pintasiri, Pongpak Chiyasak, Cristina Leonelli, Duangrudee Chaysuwan

**Affiliations:** 1Department of Engineering “Enzo Ferrari”, University of Modena and Reggio Emilia, Via P. Vivarelli 10, 41125 Modena, Italy; 2Department of Materials Engineering, Faculty of Engineering, Kasetsart University, Bangkok 10900, Thailand

**Keywords:** geopolymer, recycled corundum, alkali activation, metakaolin, mechanical performance, semireactive filler

## Abstract

Alkali activated binders, based on an aluminosilicate powder that is activated by an alkaline solution, have been proven to encapsulate a wide number of different wastes, both in the form of liquids and solids. In this study, we investigated the effect that the addition of a spent abrasive powder, mainly composed of corundum grains (RC), had on the mechanical, physical, and chemical properties of metakaolin-based geopolymers. The waste was introduced into the geopolymer matrix as a substitute for metakaolin, or added as a filler to the geopolymeric paste. The 3D cross-linking of the geopolymer structure, with and without the presence of the corundum, was investigated via Fourier transform infrared spectroscopy, X-ray diffraction, and ionic conductivity measurements of the eluate that was produced after 24 h of immersion of the sample in water. The RC powder did not significantly modify the matrix reticulation but increased densification, as observed with scanning electron microscopy, and there was increased resistance to compression by 10 wt% addition of RC, and also when added to the paste as a filler at 20 wt%.

## 1. Introduction

The term “geopolymers” was introduced by Davidovits in the 1970s to indicate a class of largely X-ray amorphous aluminosilicate materials (SiO_2_+Al_2_O_3_ content should be higher than 80%), generally synthesised at ambient or slightly higher temperatures through the reaction of a solid aluminosilicate powder with a concentrated alkaline-metal silicate and/or hydroxide solution [1]. When lower contents of SiO_2_ and Al_2_O_3_ are present in the aluminosilicate source, the literature prefers to refer to these inorganic polymer compositions as “alkali activated materials”, AAMs, or alkali activated binders or cements [2]. Nevertheless, there is still much debate in the literature on the differences between these two terms, and on the classification of these materials. Therefore, from a purely chemical point of view, talking about alkali activation in both cases is not wrong, as indicated in the books and papers cited in this manuscript. 

The use of metakaolin to produce this new class of binders, typically indicated as geopolymers, is not considered sustainable; hence, cheaper minerals, and even waste or by-products have been substituted/added to the MK-geopolymer formulations in due proportions [3,4,5]. In the present contribution, we are proposing the addition of an Al_2_O_3_-containing waste, with the idea to increase Al^+3^ content in the final 3D network of an aluminosilicate binder. With the aim to reduce the amount of waste composed of abrasive powders, we selected corundum particles that derived from erosion tests as representative samples for those technologies that adopt abrasion or erosion as one of the surface preparation steps. A range of natural and synthetic hard and inert inorganic compounds has been proposed for these scopes; once utilized, they are typically disposed of [6,7,8]. A non-exhaustive list is reported below (increasing Mohs hardness, MH):Fe_2_O_3_, hematite (recommended to polish soft ores, glass, lapidary, and soft metals, MH = 5.5–6.5);Cr_2_O_3_, chromite (recommended for samples that cannot be exposed to aluminium oxide contamination, MH = 5.5);MgO, periclase, (recommended when fine polishing wrought aluminium alloys and magnesium-based metals and alloys, MH = 5.8);SiO_2_, natural pumice (ideal for processing glass, stone, MH = 6);SiO_2_, quartz sand (multipurpose abrasive, mainly used in sandblasting, MH = 7);Al_2_O_3_, natural corundum (ideal for all types of materials and for rough as well as fine grinding, MH = 9);SiC, synthetic silicon carbide (ideal for all types of materials for rough and fine grinding and polishing, MH = 9–10);C, diamond (multipurpose abrasive, MH = 10).

Aside from automotive or naval paint sludge and other liquid waste from sand- and grit-blasting technologies, solid grit (abrasive particles) contaminated with various types of materials has been added to cement formulations since 1991 [9], with the aim of reducing the instruction of fine aggregates such as sand [10,11,12,13]. The valorization of abrasive waste powders in AAM formulations in the literature has also been reported for waste garnet abrasive powders generated from automated and manual blasting process [14,15], corundum abrasive powders [16], and corundum containing powders [17]. The observed role of these abrasive powders was, in most cases, to be that of an inert filler, even though in some cases limited reactivity was observed, specifically of the fine abrasion residues [16]. Moreover, there are some studies [18,19] that reported that reactive alumina forms when added to AAMs, exploiting the advantage of partial solutions in alkaline media.

The corundum that results from erosion tests was chosen as a representative of grit-blasting waste or spent blast abrasive, a waste category that is becoming an urgent problem for those countries that have ship repair and/or ship dismantling industries [20]. With this investigation, we intend to valorize, as a partial reactive filler, a spent abrasive corundum powder up to 20 wt% to a metakaolin-based geopolymer binder. In a previous preliminary study [16], we selected a coarse grain grit (average grain size of about 230 µm) composed mainly of corundum, Al_2_O_3_, which was substituted to MK in the mix design, and showed advantages in terms of increased mechanical resistance. With the same objective, hereafter, we presented the results that were collected for a finer spent corundum grit (52 µm) that was either added as well as substituted to the MK source; once again, metakaolin was considered to be a reference aluminosilicate powder suitable for understanding the role of the waste and its impurities during alkali activation. The aim of the present study, apart from proving the increased mechanical performance with the introduction of spent abrasive powders of corundum, was also focused on the effect of the presence of additional Al_2_O_3_ in the composition of geopolymers on the reticulation of the aluminosilicate network via (i) FT-IR spectroscopy, (ii) the determination of ionic conductivity on the leachate solution, and (iii) XRD analyses after 28 days of ageing. At the same curing time, SEM morphology, density, and finally, the compressive strength, were also determined.

## 2. Materials and Methods

### 2.1. Materials 

The metakaolin (MK) used in this study was an ARGICAL™ M1000 metakaolin, (IMERYS, France, reported chemical composition from producer as follows: SiO_2_ = 55%; Al_2_O_3_ = 40%, Fe_2_O_3_ = 1.4%; TiO_2_ = 1.5%; Na_2_O + K_2_O = 0.8%; CaO + MgO = 0.3%; LOI = 1%). This MK was used as a solid precursor for the geopolymer binders because of its high reactivity, fine particle size, and being Si/Al appropriate. 

The waste abrasive powder, also indicated as spent abrasive or waste corundum, and indicated with the acronym RC, was obtained from pure corundum (Al_2_O_3_) grains used for erosion tests on different types of materials (ceramic, glass); it was used in the as-received state. This type of waste is typically classified as EWC 12.01.17/non-hazardous waste blasting material, but there are spent abrasive powders that are also classified as EWC 12.01.16*/waste blasting material containing hazardous substances (EWC-European Waste Code, [21]). The RC’s particle size of about 50 µm, as well as the particle size distribution (D90: 81.43 µm; D50: 52 µm; D10: 34 µm) confirmed by a laser diffraction particle size analyzer (Mastersizer 2000, Malvern Instruments Ltd., Malvern, UK), are reported in Appendix A, where the powder of the as-received metakaolin is also presented. The RC can be considered to be very similar to a fine aggregate (95% particles < 75 µm, as classified by ASTM C136 - 06 Standard Test Method for Sieve Analysis of Fine and Coarse Aggregates), presenting 95% of the particles with a diameter of less than 82 µm (D95 = 82) and 100% less than 90 µm (D100 = 86). The formulations (proposed in Section 2.2) can be classified as mortars for plastering purposes, in terms of the waste corundum fineness.

The chemical composition of the RC is reported and discussed in Section 3.5.

The NaOH solution was prepared by dissolving laboratory grade granules (96 wt%, Sigma-Aldrich, Italy) into distilled water to a concentration of 8 M. The sodium silicate solution (SiO_2_/Na_2_O = 3.00 molar ratio; SiO_2_ = 26.50 wt%, Na_2_O = 8.70 wt%, and pH = 11.7), with a bulk density of 1.34 at 20 °C, was provided by Ingessil, Verona, Italy. Hereafter, we refer to this solution as NaSilicate.

Since one of the objectives of this study was to evidence the eventual effects of Al_2_O_3_ presence in geopolymer 3D networks, prior to the introduction of recycled corundum powder to the geopolymers, the ability to dissolve corundum in a basic environment was studied. As seen in several papers [22,23], dissolution is directly proportional to the adsorption of the OH- ions. Moreover, for this reason, an 8 M NaOH solution was chosen. As explained by Kumar et al., 2015 [19], the size of a material affects the mechanical properties, the dissolution in both basic and acid environments, and the cross-linking that may occur when reacting. Furthermore, the dissolution of aluminium oxides also depends on the temperature [24]; in fact, we did not expect a high reactivity from the recycled corundum powder, both for its particle size and for having worked at room temperature.

### 2.2. Preparation of Geopolymer Specimens

The geopolymers were prepared by adding 38 g of NaOH to 100 g of MK, thus starting the alkali activation; then, 40 g of sodium silicate solution was added under mechanical stirring to obtain the reference geopolymer, GP0 (Table 1). To investigate the possibility of substituting the as-received waste to MK, RC replaced MK by 10% and 20% by weight (measured on the dry powders) to produce the geopolymer mortars labelled GP-10RC and GP-20RC (Table 1). The percentages of RC on the total amount of fresh paste accounted for 5.88 and 12.84 wt%, respectively. Different formulations were obtained by adding 10% and 20% by weight of corundum to the geopolymer fresh paste (i.e., the formulation GP0), labelled 90GP0-10RC and 80GP0-20RC, respectively (Table 1). In these two mixes, the percentages of RC on the total amount of fresh paste accounted for 10% and 20 wt%, respectively. For all of the compositions, a constant alkali activator solution (NaOH + NaSilicate = L)/MK weight ratio of 0.765–0.780 was maintained. All of the pastes were prepared with an Aucma 1400W Planetary Mixer (China). 

The fresh binder paste was poured into silicone cube moulds (25 mm × 25 mm × 25 mm). The geopolymer pastes that were obtained from the various samples in this study were quite viscous, and no water was added to any formulation. In Table 1, it is possible to note that the liquid (8 M NaOH plus sodium silicate solution) to solid (MK and RC) mass ratio remained in the range 0.54–0.78, being higher for the higher MK content, assuring the proper workability of all pastes. The impervious RC grains increased the flowability of the paste, as fine filler usually does, and necessitated minor water addition. After removing all bubbles, the moulds were carefully closed, and the geopolymers were cured at room temperature. The silicone moulds were opened after 28 days of hardening, in order to proceed with the correct characterisation. A minimum of 8 samples were obtained for each formulation.

### 2.3. Geopolymers Characterisation

#### 2.3.1. Reactivity Test in NaOH Solution

The basic attack method was used to quantify the fraction of potentially reactive material. Recycled corundum was checked for its reactivity in an alkaline environment, after immersion in NaOH solution. The basic attack was made to reproduce the condition of alkali activation. The reaction took place in a flask that was wetted at 80 °C, and the solution used in the etching was 8 M NaOH. The test was carried out by inserting 1 g of RC in 100 mL of 8 M NaOH solution. Then, it was stirred constantly for 5 h. Once the test was finished, the solution was filtered, and the residue was washed with milliQ water to then characterise the powder through X-ray diffraction (X’Pert PRO, PANAlytical, Mal-vern Panalyical Ltd., Malvern, UK). The NaOH solution was prepared by dissolving laboratory-grade granules (96 wt%, Sigma-Aldrich, Milano, Italy) into distilled water to reach a concentration of 8 M. 

#### 2.3.2. Microstructural Characterisation 

Fourier transform infrared spectroscopy, FT-IR, (Avatar 330 FTIR, Thermo Nicolet, White Bear Lake, Minnesota, USA) was performed on the powder of each sample. A minimum of 32 scans between 4000 and 500 cm^−1^ were averaged for each spectrum at a 1 cm^−1^ interval [25]. 

#### 2.3.3. Ionic Conductivity of the Geopolymer Leachate

Ion conductivity measurements were performed to observe the chemical stability of the cross-linked 3D aluminosilicate network of the prepared geopolymers. The ionic conductivity of all of the samples was measured with a Crison GLP31 (Hach Lange Spain, SLU, Barcelona, Spain), and we compared these results with the ionic conductivity of MilliQ water. MilliQ water (1:10 solid-water ratio) was added to the ground powder geopolymer samples. Ionic conductivity measurements were performed at different times: t1 = 0 h, t2 = 0.2, t3 = 0.25, t4 = 0.5, t5 = 1, t6 = 2 h, t7 = 8 h, t8 = 24, and t9 = 48 h.

#### 2.3.4. Mineralogical Composition

Mineralogical analysis of the powders (metakaolin and corundum waste) and consolidated geopolymers were carried out with an X-ray powder diffractometer, XRD, (PW3710, Phillips, UK) CuKα, Ni-filtered radiation (the wavelength was 1.54184 Å). Diffraction patterns were collected by the X’Celerator detector, from 5 to 70° 2θ, with a step size of 0.02° 2θ, and a counting time of 2 s. Mineral phases were identified by comparing the experimental peaks with reference patterns (DIFFRAC plus EVA software, 2005 PDF2, Bruker, Billerica, MA, USA).

#### 2.3.5. Microstructural SEM Observation

Microstructural analyses were performed on 28 days consolidated mortars, where the specimens for each of the examined preparations were tested after the compressive strength test. For morphology and microstructure investigations, scanning electron microscopy (SEM) (FEI, Quanta 450, Pisa, Italy) analysis coupled with chemical analysis by energy dispersive spectroscopy (EDS) were performed; the specimens were sputtered with gold (Au) for electrical conductivity, and with the intent of capturing better displays [25,26]. For the instrument conditions of SEM images, an accelerating voltage of 10 kV was applied, using magnifications of 500X and 1000X.

#### 2.3.6. Physical and Mechanical Properties 

The true density, as the weight of fine geopolymer powder without air in its open pores, was determined with the help of a pycnometer at room temperature (25 °C) [27,28]. The pycnometer was filled with liquid (*w*_3_), and the dry pycnometer and stopper (*w*_0_) were weighed on an analytical balance. The geopolymer powder was inserted into the dry pycnometer (*w*_1_), which was then filled fully with water, stoppered, and boiled to remove the air bubbles. The outside was dried, and the full pycnometer was again weighed on the analytical balance (*w*_2_). The true density (g/cm^3^) was calculated from Equation (1):(1)$$\mathrm{True}\;\mathrm{density}=\frac{w_{1}-w_{0}}{(w_{3}-w_{o})-\left(w_{2}-w_{1}\right)}\left(D_{L}-D_{A}\right)+D_{A}$$
where D_A_ = density of air (1.184 × 10^−3^ g/cm^3^) [29], and D_L_ = density of water (1.000 g/cm^3^) [27,29].

To test the mechanical properties of the geopolymers, compression tests were performed with an Instron 5567 Universal Testing Machine (Norwood, MA, USA) after 28 days of curing. The load (30 kN load limit) was applied and increased at a displacement rate of 1 mm/min. The tests were executed in displacement control mode at a constant loading velocity with no preload. They were stopped after obtaining eight valid tests for each different geopolymer composition. The compressive strength values are the mean value of eight tests.

## 3. Results

### 3.1. Reactivity of Waste Corundum Powders in NaOH

Figure 1 shows the XRD spectra of pure alumina and fine corundum powder after being tested in 8 M NaOH. The pure alumina spectra after the basic attack were inserted simply to accentuate the differences between the two powders after the reaction in NaOH. In fact, alumina powder is not affected in any way by the presence of sodium hydroxide, while spent corundum powder can also be seen by comparing the spectrum in Figure 1 to those of RC before alkali attack, as reported in Figure 4. It was observed that in RC before the attack, there were traces of contaminants from eroded materials, namely montmorillonite (Na,Ca)_0.3_(Al,Mg)_2_Si_4_O_10_(OH)_2_·nH_2_O and valleriite 2[(Fe,Cu)S]·1.53[(Mg,Al)(OH)_2_], which were not present after the basic attack; moreover, for this reason it was decided to use NaOH at this specific concentration.

### 3.2. Microstructural Characterisation

Figure 2 reports the FT-IR spectra of MK, RC powders, and the composite geopolymers GP-RC from 0 wt% to 20 wt%. At around 3440 and 1660 cm^−1^ in all spectra, we can see -OH stretching and bending. The MK spectrum shows the Si-O-T bands (T = Si or Al) at 1080 cm^−1^. Indeed, this band shifts to a lower wavenumber (1019–1014 cm^−1^), suggesting increased Si-O-Al bonds. In the MK spectrum, the 800 cm^−1^ band is due to the presence of quartz, while the 560 cm^−1^ band is related to Al-O vibration in six-fold coordination. The band at 470 cm^−1^ in MK and in GP (from 0 to 20 wt% of RC) was assigned to Si-OH bending mode (see Table 2 and the comments on the peak at 460 cm^−1^ for RC).

### 3.3. Ionic Conductivity of Leachate Solution

The ionic conductivity, IC, was studied to obtain structural information on the chemical stability of the geopolymer, to which a waste material was added both in the precursor and as a filler. In contrast to vibrational spectroscopies (FT-IR and RAMAN), this type of investigation is considered to provide an indirect indication of the reticulation degree of the aluminosilicate matrix. The higher amount of Al^+3^ is found in tetrahedral coordination, the lower the counter-ion, Na^+^, leaching is registered. In addition, the higher the OH- ions from alkali activator reacting with aluminosilicate precursors, the lower the leaching is. In conclusion, the ionic conductivity of the eluate solution obtained after the geopolymer immersion is a quantitative measure of its reticulation degree. 

In the formulations under investigation, IC increases with the addition of corundum powder to MK powder, as expected by a non-reactive or partially reactive mix component (Figure 3A). It can be noted that the difference between GP0 and the geopolymer was relevant for the formulations from GP-10RC to 80GP0-20RC, while the formulation with 10% of RC as filler was closer to GP0. It also continued to rise with the addition of RC to the geopolymer paste (GP0); however, obviously when introduced as a filler after about 24 h, samples 90GP0-10RC and 80GP0-20RC began to have similar values. Similar values of IC for these compositions indicated that the reticulation of the geopolymeric paste was almost not affected by the addition of RC. When RC was added to the MK powder, the values of IC were lower than in the previous case, due to the presence of Al_2_O_3_ before alkali activation. The general trend of the ionic conductivity can be related to the overall content of alkali activator (8 M NaOH plus Na silicate) with a good inverse proportionality, as reported in Figure 3B. The higher the alkaline solution, the lower the release of ions in the water of the test during immersion. The good proportionality (R^2^ = 0.9794 insert value after correcting IC data in the plot) indicated that the reactivity of the solid fraction, MK and RC, increased with an increase in liquid activator, and also coincidentally with an increase in the MK content. The presence of the RC slightly affected the reticulation of the geopolymeric matrix, following the same curing process for all of the formulations. Furthermore, when the two solid components were separately correlated with IC (Figure 3C,D), the same correlation was found for RC (R^2^ = 0.9571), as well as for MK (R^2^ = 0.9547). These results indicated that the mix design was properly executed, allowing for maximum reactivity and OH- uptake from part of the reactive fraction of the formulation, i.e., MK.

### 3.4. Mineralogical Composition

Figure 4A compares the spectra of recycled corundum and metakaolin powders with GP-10RC and GP-20RC. In the diffraction pattern from metakaolin, there was evident diffuse reflection identified as the typical large band of an amorphous aluminosilicate structure, plus shaper peaks that were identified as anatase (TiO_2_) and alpha-quartz (α-SiO_2_). The diffraction patterns of GP-10RC and of GP-20RC were similar (Figure 4A), and these two geopolymers had a diffuse halo characteristic of amorphous aluminosilicate networks shifted toward higher angles, i.e., at about 26–28° in 2θ. Such a shift has been noted in MK-based geopolymers with the 3D reticulation of Si- and Al -tetrahedra. As the percentage of RC increased, a peak of about 8° in 2θ began to be noted in the sample GP-20RC, which confirmed the presence of recycled corundum which did not react adequately. This was not noticeable in the GP-10RC sample’s spectrum, due to the smaller amount of RC in it. In Figure 4B is shown a comparison between MK, RC powders, and the geopolymers, with RC added as a filler. The diffraction patterns of 90GP0-10RC and 80GP0-20RC also showed diffuse halos that are characteristic of amorphous geopolymers [32]. With the addition of the RC, it is also possible to note that in the spectra there is an absence of peaks at 8° and around 14° in 2θ, due to the impurities present in the recycled corundum. The absence of such peaks is an indication that the alkaline environment was basic enough for impurities of the waste RC to react and dissolve; meanwhile, corundum (α-Al_2_O_3_) peaks are still clearly visible.

### 3.5. Microstructural SEM Observation

As shown in Figure 5A, the microstructure of waste corundum grains was observed to be prismatic in shape, with sizes in the range of 50–500 µm. Figure 5B shows a finer powdery deposit at the surface of the corundum grains that can be attributed either to finer Al_2_O_3_ particles, or to contaminants from the erosion operation. As far as the chemical composition of the waste is concerned (Figure 5C), it appeared rich in oxygen (O) (52 ± 6 wt%), aluminium (Al) (32 ± 5 wt%), with a good approximation to an Al/O atomic ratio equal to 0.62 (with respect to a 0.67 theoretical ratio); this indicated that there was a high content of Al_2_O_3_, with a relatively low content of contaminants deriving from the erosion testing. In particular, we recorded sodium (Na) (1.62 wt%), nitrogen, Ti, Fe, Ti, Fe, and Si, consistent with the crystalline impurities of aluminosilicates found from XRD diffraction (Figure 4A,B). The presence of considerable contents of carbon (8.55 wt%) was due to the presence of carbon adhesive tape that was used for sample preparation.

However, as the SEM analyses were performed after the compressive strength test, the presence of cracks was observed at the surface of the specimens. Figure 6 shows the morphology of the geopolymers and the unreacted waste corundum grains embedded in the matrix. Two phases were distinguished: (i) the unreacted waste corundum, or partially reacted particles; and (ii) sponge-like geopolymer gels, denoted by “A” and “B,” respectively. More geopolymer gels could be detected at higher waste corundum contents, indicating that a higher degree of geopolymerization had occurred [33]. From Figure 6, it is possible to observe that specimen 90GP0-10RC was more compact than the other formulations, which led to the highest compressive strength among the geopolymer specimens. 80GP0-20RC had a higher amount of waste corundum than other formulations, which led to a higher aluminium content in its chemical composition, combined with a higher amount of geopolymer gel. The chemical compositions, as analysed via EDS for the geopolymers, were as follows: GP-10RC (C: 9.82; Na: 12.04; Al: 29.49; Si: 45.57; K: 0.55; Ca: 0.13; Fe: 2.40 wt%), GP-20RC (C: 9.36; Na: 12.46; Al: 29.32; Si: 45.27; K:0.83; Ca: 0.23; Fe: 2.53 wt%), 90GP0-10RC (C: 10.29; Na: 12.42; Al: 30.17; Si: 44.29; K: 0.70; Ca: 0.23; Fe: 1.90 wt%), and 80GP0-20RC (C: 11.48; Na: 13.64; Al: 28.90; Si: 43.34; K: 0.67; Ca: 0.16; Fe: 1.81 wt%). The error was about 5 to 15%.

### 3.6. Physical and Mechanical Properties

Figure 7 presents the trend of the density values correlated with those for the compressive strength of all geopolymers tested. The density was calculated accordingly with Equation (1), and for GP0 it was found to remain at around 1.8–2.2 g/cm^3^, as reported in the literature for metakaolin-based geopolymers [34,35,36]. The density values of the geopolymers are slightly higher than that of the GP0, to which was added a denser filler, RC. The experimental value found for spent corundum was 3.83 ± 0.27 g/cm^3^, which was slightly lower than the density of pure corundum, 2.94–4.1 g/cm^3^ [37,38]. It is noted that the density slightly increased, as a result of the addition of RC to MK of 10 to that of 20 wt%, before adding the alkali activator in GP0-10RC and GP0-20RC. A similar trend was found for 90GP0-10RC and 80GP0-20RC, when the corundum waste was added to the fresh geopolymeric paste. The trend of the density values is in good agreement with the expected data, theoretical densities, which were calculated from the mixture rule applied to the geopolymer formulations, in consideration of the real RC content as from Table 1. The slight differences between experimental density and theoretical density fall within the variability of the samples that produced an error of 7% on the density values. It can be noticed that the densification of the geopolymer is higher for the RC added as a filler, the experimental value being higher than theoretical one with respect to formulation with RC substituted to RC independently upon the overall RC fraction. Such a tendency reflects the SEM observation where a higher densification of the matrix was also evidenced.

Concerning the compressive strength (Figure 7), it can be seen that after the addition of recycled corundum in MK, the mechanical properties increased slightly; however, in any case, the values decreased a lot compared to the reference geopolymer GP0, as was already discussed in the ionic conductivity paragraph. By adding 10% by weight of RC as filler instead, therefore, after the alkali activation already occurred, the value of the compressive resistance increased. On the other hand, when 20% was added, the resistance decreased considerably, which was also mentioned in the paragraph on ionic conductivity; this was due to the excessive presence of RC compared to the reacted geopolymeric paste. It should be noticed that the GP-20RC sample had an actual RC fraction of 12.34 wt% of the total paste weight (Table 1), which was different from the actual fraction of 20% for 80GP-20RC.

The difference between adding RC to MK or adding RC to the paste as a filler is more evident in the mechanical properties (Figure 7) than in ionic conductivity (Figure 3). In the case of the GP-10RC and GP-20RC formulations, the higher the conductivity, the higher the compressive strength. For the 90GP0-10RC and 80GP0-20RC samples, on the other hand, the ionic conductivity values were similar after 48 h, but the compressive strength varied a lot. 

## 4. Discussion

Corundum-based abrasives are available in many grain size fractions, with different mineralogical and chemical purities [39,40]. Corundum is chemically and thermally inert; thus, it has potentially good recycling possibilities if the contamination from the abraded or eroded materials does not carry too many hazardous elements. Checking for the presence of these contaminants was one of our priorities.

In this study, it was proven via XRD (Figure 1) and EDS (Figure 5) that the waste abrasive powder collected after the erosion testing procedure did contain only α-Al_2_O_3_ corundum, with minor phases that are typical of ceramic materials, of montmorillonite, and of metals such as valleriite. In contrast with pure corundum (Figure 1), the spent abrasive powder is capable of some reactivity in the hot NaOH reactivity test, preferably with the contaminant aluminosilicate phases generating a zeolite-like phase, which are typical of alkaline aluminosilicate solutions above room temperature [41]. From these results, we expected some reactivity of the fine RC powder in the alkaline environment; thus, we attempted to produce metakaolin-based geopolymer formulations using concentrated solutions of NaOH and sodium silicate. Fineness of the RC powder, with grain sizes less than 80 μm (average value or D50 was 52 μm via laser scattering grain sizer, and confirmed via SEM), was not sufficient to dissolve the corundum phase, as proven by the FT-IR (Figure 2) and XRD (Figure 4) investigations; nevertheless, we assisted the dissolution of the contaminants, which fostered the production of a sponge-like geopolymer gel derived from MK dissolution and reticulation, as is visible in SEM images (Figure 6).

The reticulation of the 3D aluminosilicate geopolymer network was not hindered by the presence of the RC powder up to the maximum test addition of 20% (Table 1), as indicated by the peak positions in the FT-IR spectra shown in Figure 2. From the density measurements, we confirmed that the reticulation of the MK regularly occurred in all of our formulations, given the alkaline solution to MK weight ratio remained constant at approximately 0.77 (namely in the range of 0.765–0.780; see Table 1). The experimentally determined density values were in line with the theoretical values (Figure 7), where the theoretical data were calculated as though the reticulations of all of the samples were the same as in the GP0 formulation. The most sensitive, yet indirect, analytical approach we adopted to determine the degree of geopolymer reticulation was the measurement of the ionic conductivity of the eluate, after the immersion in distilled water of a piece of consolidated geopolymer [42]. As stated in Section 3.3, an increase in the ionic conductivity is proportional to the overall amount of the liquid activator, while it is completely independent of the amount of the RC (Figure 3).

Following the same curing process for all of the formulations, hence presenting the same type of geopolymeric matrix in all of the formulations generated by the MK rather than by the RC, the role of corundum grains remained as that of an inert filler. An increase in compressive strength was recorded for the formulations that contained around 10 wt% of RC, and preferably in the case when the mix design proposed the addition of RC after the proper mixing of MK plus the alkaline solutions. In this case, the intimate contact of MK lamellas and the activator induced a better reticulation that developed better bonding with RC grains [43].

## 5. Conclusions

The recycle/reuse of spent corundum has seldom been investigated in the literature [40], yet inclusion of inert wastes in geopolymer matrices is a widely shared approach [44]. The formulations proposed in this study presented a good degree of reticulation, even in presence of spent corundum grit up to a maximum of 20 wt%, calculated on the basis of fresh paste. The presence of the waste increased the compressive strength resistance of the geopolymeric matrix. Our formulations should be considered performant when compared to literature data, where copper slags used as grit for blasting had been added to OPC in ratios by weight of 1:1, 1:3, and 1:4 [9]. In that study, the compressive strength values at 28 days were 35, 34, and 33 MPa, respectively, when tested on mortar cubes of size 25.4 × 25.4 × 25.4 mm^3^. Considering that the RC is not as reactive as copper slag, the values that were recorded in this study are comparably high.

## Figures and Tables

**Figure 1 polymers-14-05091-f001:**
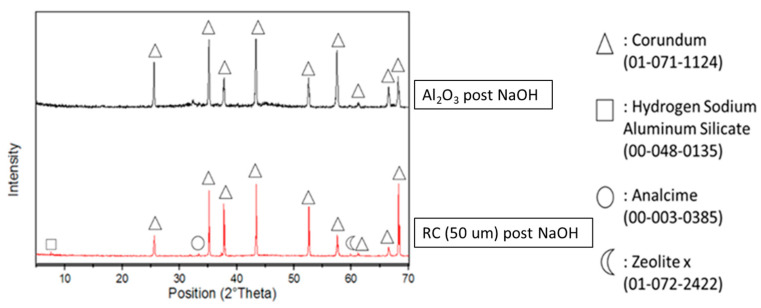
XRD patterns for alumina and RC, post-attack with 8 M NaOH.

**Figure 2 polymers-14-05091-f002:**
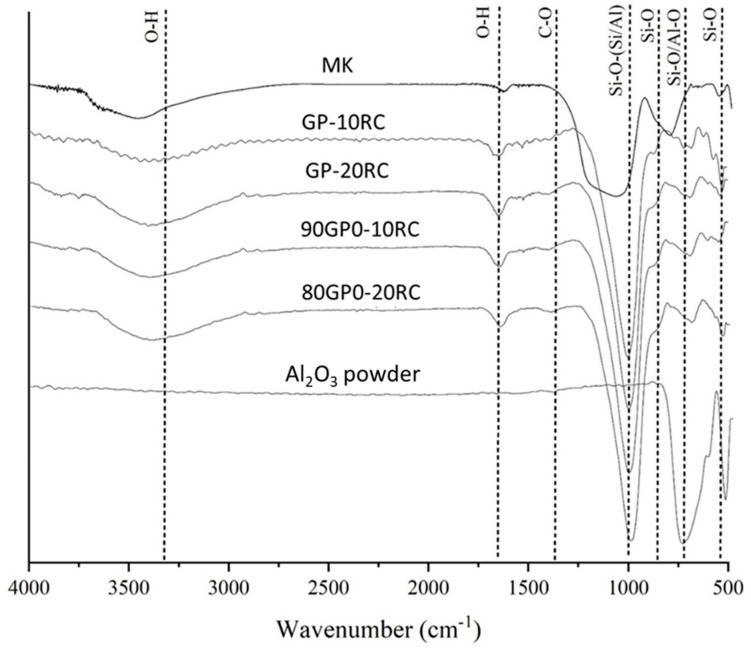
FT-IR spectra of MK powder, all geopolymers with Al_2_O_3_, and Al_2_O_3_ powder.

**Figure 3 polymers-14-05091-f003:**
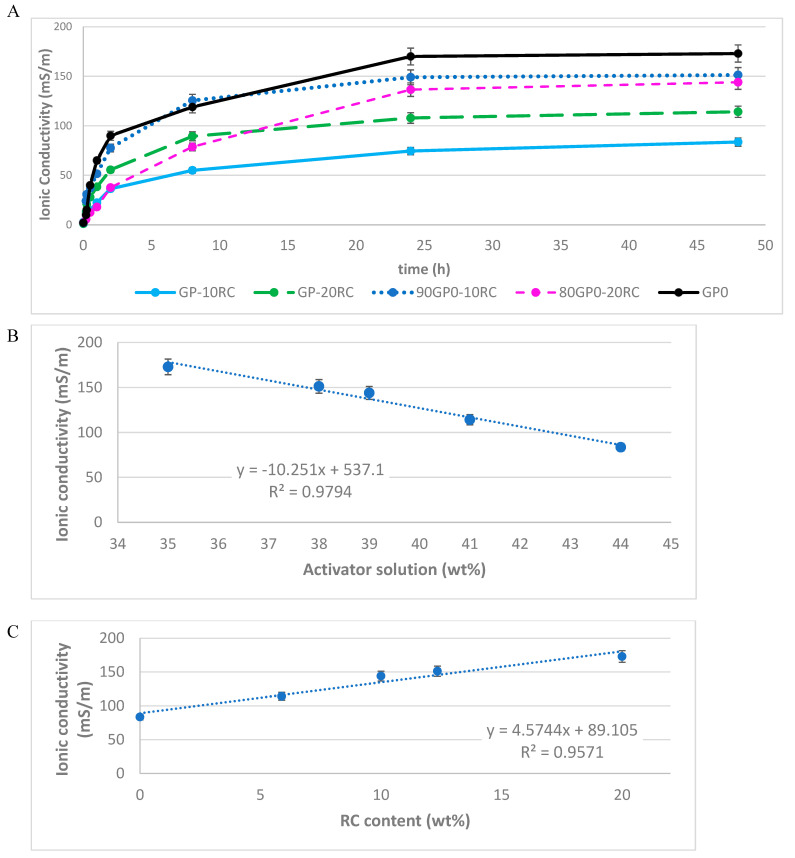
Ionic conductivity (IC) at 48 h of immersion vs. (**A**) immersion time of all geopolymers cured 28 days; (**B**) IC vs. the alkaline solution content; (**C**) IC vs. RC content; (**D**) IC vs. MK content. (Reproducibility of the test was calculated to be within an error of ±5%).

**Figure 4 polymers-14-05091-f004:**
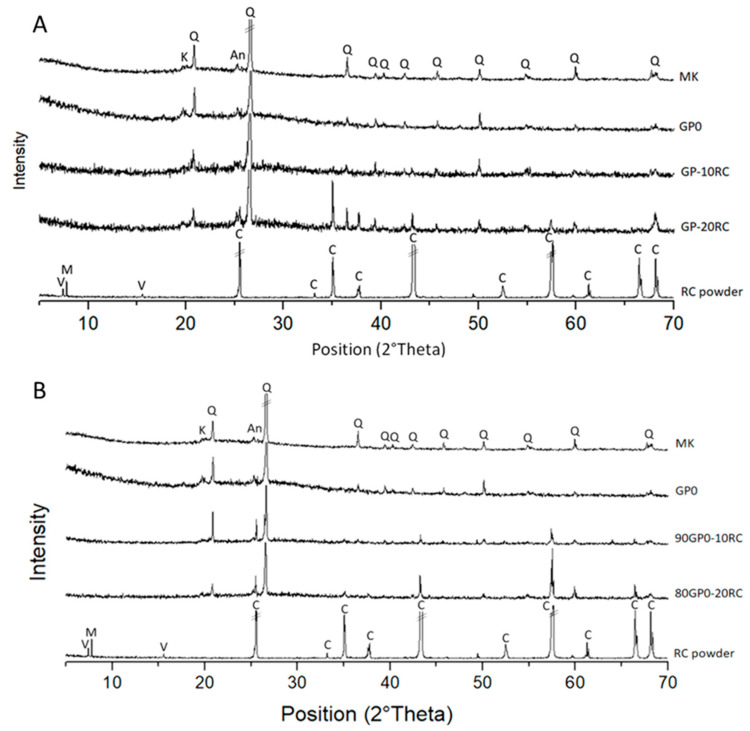
(**A**) XRD patterns for RC raw materials MK, GP-10RC, and GP-20RC recycled corundum-based geopolymers. (**B**) XRD patterns for RC raw materials MK, 90GP0-10RC, and 80GP0-20RC. Crystalline phases identification labels: Q = alpha-quartz (01-083-0539), K = kaolinite (01-075-0938), C = corundum (01-071-1124) An = anatase (01-071-1167), M: montmorillonite (00-012-0232), V: valleriite (01-073-0516).

**Figure 5 polymers-14-05091-f005:**
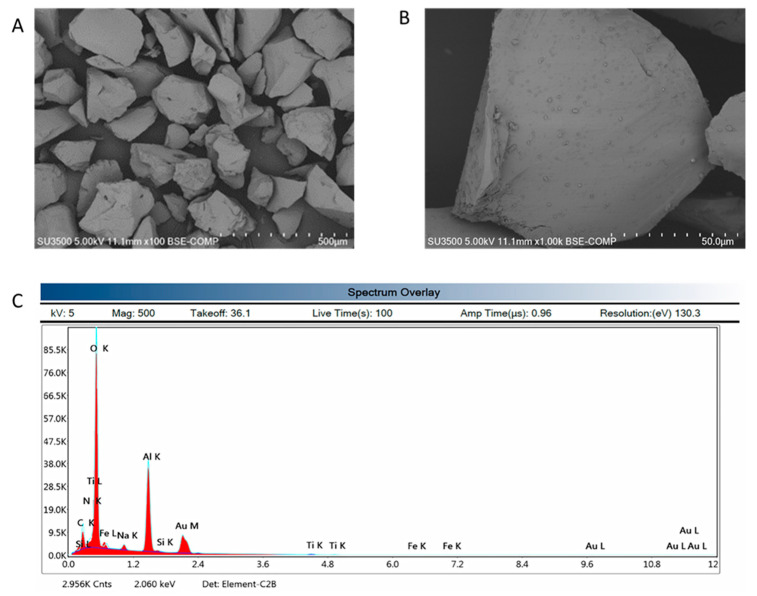
Microstructure of waste corundum via SEM (**A**) at 500X, (**B**) at 1000X, and (**C**) chemical analyses via EDS.

**Figure 6 polymers-14-05091-f006:**
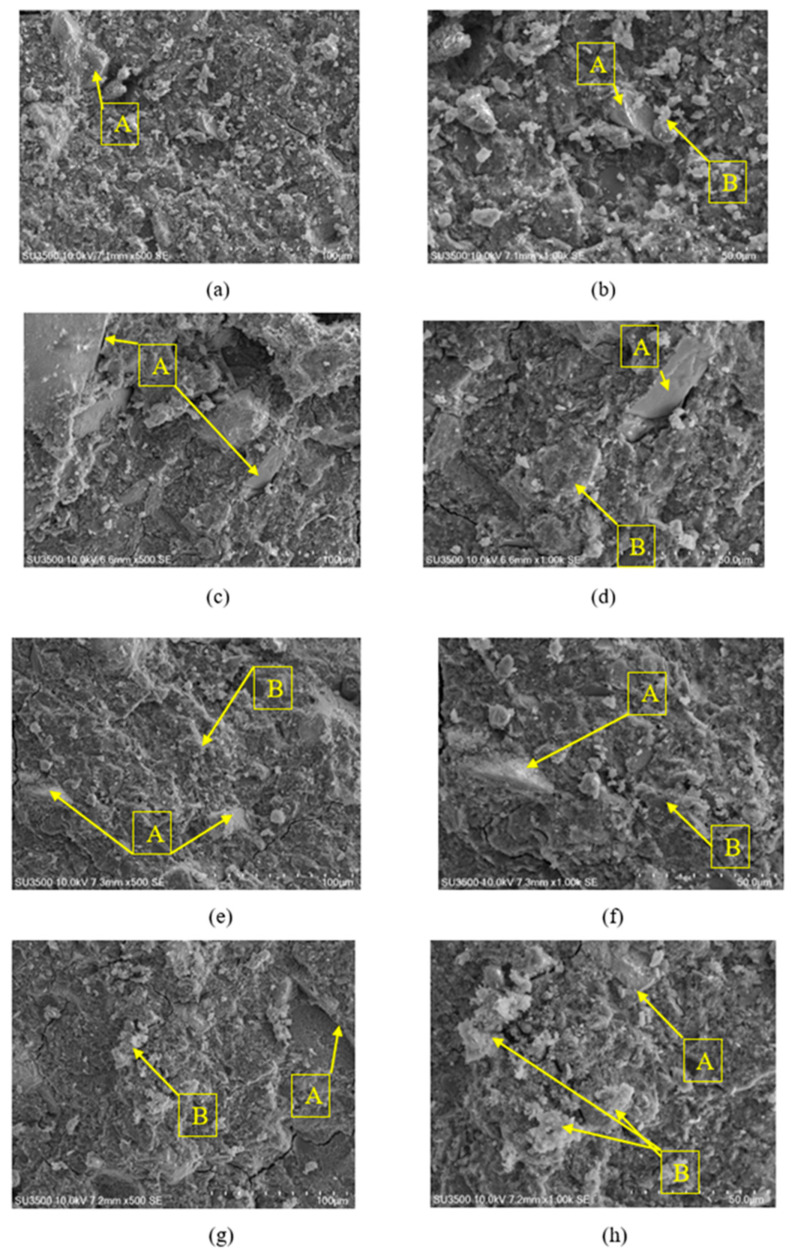
Microstructures of fresh surfaces of the different mortars: (**a**) GP-10RC, 500X; (**b**) GP-10RC, 1000X; (**c**) GP-20RC, 500X; (**d**) GP-20RC, 1000X; (**e**) 90GP0-10RC, 500X; (**f**) 90GP0-10RC, 1000X; (**g**) 80GP0-20RC, 500X; and (**h**) 80GP0-20RC, 1000X. Unreacted waste corundum or partially reacted particles and sponge-like geopolymer gels are denoted by “A” and “B”, respectively.

**Figure 7 polymers-14-05091-f007:**
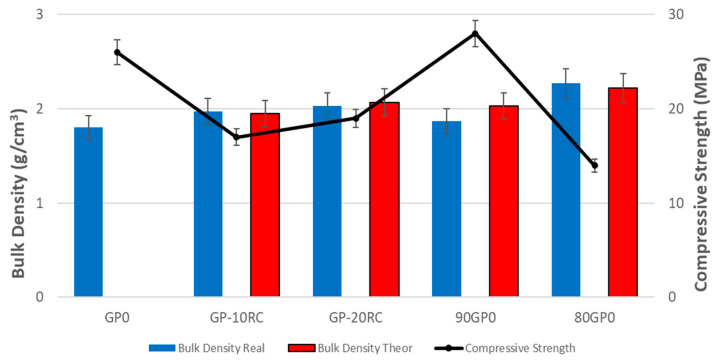
Compressive strength compared with density (theoretical values and calculated values) for all of the geopolymers after 28 days of curing time.

**Table 1 polymers-14-05091-t001:** Mix design of mortar formulations containing various amounts of RC. L/S indicates the liquid to solid wt/wt ratio, where liquid has to be the intended 8 M NaOH plus sodium silicate solution. L/MK wt/wt ratio indicates the liquid to MK wt/wt ratio, where L = NaOH + NaSilicate (See text for more details).

Mix ID	MK (g)	RC Powder (wt%)	NaOH (g)	NaSilicate (g)	L/MK (wt/wt)	L/S (wt/wt)	SiO_2_ (wt%)	Na_2_O (wt%)	Al_2_O_3_ (wt%)	H_2_O (wt%)	Other Oxides (wt%)
GP0	100	0	38	40	0.78	0.78	37	6	22	32	3
GP-10RC	90	5.88	34	36	0.778	0.7	35	6	27	30	2
GP-20RC	80	12.34	30	32	0.775	0.62	33	5	31	28	3
90GP0-10RC	51	10	19	20	0.765	0.64	34	5	30	28	3
80GP0-20RC	45	20	17	18	0.778	0.54	30	5	36	25	4

**Table 2 polymers-14-05091-t002:** List of vibrations of aluminosilicates and corundum in the FT-IR range.

Wavenumber (cm^−1^)	Interpretation	Reference
3695−3660 cm^−1^	Hydroxyl groups in the kaolinite	[25,30,31]
3450 and 1650 cm^−1^	Hydration water	[30,31]
1454 cm^−1^	Na_2_CO_3_	[31]
1080−1050 cm^−1^	Asymmetrical Si−O−Si Stretching vibration	[25,30]
1010−980 cm^−1^	Si−O−Al stretching vibration	[16,30,31]
912 cm^−1^	Al (VI)−OH stretching	[30,31]
800−500 cm^−1^	Al−O−Si bending vibrations	[30,31]
450−470 cm^−1^	Si−O−Si bending vibration	[16,25,30,31]
1089 cm^−1^	Al−O vibration	[16,31]
1089, 800 and 780 cm^−1^	Disorders in Corundum network	[16]
459 cm^−1^	Al−O bending	[16,30,31]
474 cm^−1^	Al−O vibration Al_2_O_3_	[16]

## Data Availability

Not applicable.

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
