# Peer review of "Chemical and Mechanical Properties of Metakaolin-Based Geopolymers with Waste Corundum Powder Resulting from Erosion Testing"

_polymers, 2022, doi:10.3390/polym14235091_

Round 1
Reviewer 1 Report
Chemical and mechanical properties of MK-based geopolymers 2 with waste corundum powder resulting from erosion testing
Abstract:
• Components present but may need minor writing for readability
• Subject matter is original and important
Introduction:
Minor rewriting; the hypothesis is not clearly presented.
• References are adequate.
Remarks:
Chemical forms: (SiO2+Al2O) can be written as (SiO2+Al2O).
The hypothesis and objective are not clearly stated, please add them at the end of the introduction.
WHY what is your contribution?
Materials and Methods:
Adequately written, although writing could be polished; minor typos/grammar/ punctuation errors
• Description of procedures needs minor clarification (clearly remediable)
Remark:
• Statistical significance of the data and measurements are not stated.
Results:
Minor rewriting is needed to support all items mentioned in Materials and method.
Remark:
Improve fig.6. scale and resolution.
Discussion:
Statements, goals, and conclusions could be more supported by data; rewriting could address this
“Why/How/What, significant results should be highlighted”
• Some study implications and/or limitations are missing or not clearly presented
• Study can advance knowledge if the paper is rewritten and key components clearly presented.
Reviewer 2 Report
This paper aims to the feasibility of using waste corundum powder in MK-based geopolymers. The content of the article is comprehensive, but some details are not well done. There are some comments below that authors may consider revising their paper.
1. In the introductory part, there is insufficient research on the previous literature. In addition, the shortcomings of previous studies and the necessity of this study should be added.
2. 2.1, What is the chemical composition of waste corundum powder? A table containing the chemical composition of MK and waste corundum powder is required here.
3. What is the difference between GP-10RC, GP-20RC and 90GP0-10RC, 80GP0-20RC? Also, what is the reason for this design?
4. L88, The meaning of high reactivity and fine particle size needs to be specified.
5. L89, “waste abrasive powder (RC)” Is it corundum waste powder?
6. How much sodium silicate is used?
7. What is "L/S" in Table 1?
8. Are there any references for equation (1)?
9. The discussion part should analyze and discuss the mechanism based on the experimental results, rather than simply listing the experimental results.
10. No conclusion section?
Reviewer 3 Report
The paper presents the results of experimental research in the area of growing interest connected with the use of wastes and by-products for new materials. The alkali-activated materials (AAM) – in this number geopolymers, are considered an excellent option thanks to their ability of the immobilization of the environmentally hazardous components of the wastes.
It is certainly an interesting attempt to show other possibilities for the use AAM. This area is of certain importance for researchers seeking new environmentally friendly construction materials. This original paper, referring own research, addresses a certain gap in this field.
The research part is well-preceded by the Introduction based on a well-structured and sufficient state-of-art report.
The research is well-designed and executed, the methodology is carefully and clearly explained, and the results are well-described.
There are several editorial issues in the text, as for example following:
Abbreviations used in the title (MK) and abstract (FT-IR, XRD, SEM) are not revealed before the use of the abbreviation. It should be corrected.
Chemical compounds should be noted due to the rules, i.e., with underscripts (SiO2, Al2O, etc.).
Notations as those in the tile of Figure 4: (Na,Ca)0.3(Al,Mg)2Si4O10(OH)2·nH2O 00-012-0232) makes the reading difficult and shall not be encountered in the text submitted to the scientific journal.
The same problem appears with the units (e.g., v. 160: cm-1, v. 193, 343, 346…: cm3 and further, many of such mistakes in the text – power -1, 3 should be noted with the use of the superscript). The whole text shall be revisited, and those notations have to be corrected accordingly.
Figure 6: XRD record photos could be of larger scale in the text; the details are barely visible.
Figure 7: there is no reason for the word “strength” to be started with a capital letter
There are no conclusions and the discussion is very brief and do not refer to qualitative but also not really quantitative arguments. This part of the text shall be revisited and extensively corrected. The authors do not draw a line to potential future directions of research based on the one described in this research. This could certainly be an important input as well.
The paper is well-written in English, draws and tables are clear to the reader. The paper is edited on a good level.
Reviewer 4 Report
1. Line 35 “…to increase the Al+3 con- 35 tent in the final 3D network…” : put +3 in superscript. Same remark for the all document
2. Putting numbers into indices: Al2O3 =>Al2O3. Same remark for the all document
3. In the document you don’t differentiate between geopolymer and activated alkali materials. Vocabulary needs to be harmonized throughout the document
4. Can you give the volumes of Corundum produced each year in Italy and in the world?
5. You need to make a comparison of your results with the results of the literature, where waste has replaced metakaolins, for example:
- Bouchikhi, Abdelhadi, et al. "Use of residual waste glass in an alkali-activated binder–Structural characterization, environmental leaching behavior and comparison of reactivity." Journal of Building Engineering 34 (2021): 101903.
6. The English of the document needs to be improved
7. Some results are missing for the interpretation of the conclusions to be complete (total porosity, porosity distribution, TGA, etc.)
Round 2
Reviewer 2 Report
Table 1 is still difficult to understand. Does GP-10RC mean 90%MK and 10%RC? Why is the RC percentage in Table 1 5.88%? In 90GP0-10RC, MK is 51g and RC is 10%? Set the amount of raw material added to a unified unit, g or %.
Reviewer 3 Report
Thank you for your responses and extensive corrections.
